# UGSL: A Unified Framework for Benchmarking Graph Structure Learning

## Abstract

Graph neural networks (GNNs) demonstrate outstanding performance in a broad range of applications. While the majority of GNN applications assume that a graph structure is given, some recent methods substantially expanded the applicability of GNNs by showing that they may be effective even when no graph structure is explicitly provided. The GNN parameters and a graph structure are jointly learned. Previous studies adopt different experimentation setups, making it difficult to compare their merits. In this paper, we propose a benchmarking strategy for graph structure learning using a unified framework. Our framework, called Unified Graph Structure Learning (UGSL), reformulates existing models into a single model. We implement a wide range of existing models in our framework and conduct extensive analyses of the effectiveness of different components in the framework. Our results provide a clear and concise understanding of the different methods in this area as well as their strengths and weaknesses.

## 1 Introduction

Graph Representation Learning (GRL) is a rapidly-growing field applicable in domains where data can be represented as a graph (Chami et al., 2022). The allure of GRL models is both obvious and well deserved – there are many examples in the literature where graph structure information can greatly increase task performance (especially when labeled data is scarce) (Perozzi et al., 2014; Abu-El-Haija et al., 2019). However, recent results show that the success of graph-aware machine learning models, such as Graph Neural Networks (GNNs), is limited by the quality of the input graph structure (Palowitch et al., 2022). In fact, when the graph structure does not provide an appropriate inductive bias for the task, GRL methods can perform worse than similar models without graph information (Chami et al., 2022).

As a result, the field of Graph Structure Learning (GSL) has emerged to investigate the design and creation of optimal graph structures to aid in graph representation learning tasks. The relational biases found through GSL typically use multiple different sources of information and can offer significant improvements over the kinds of 'in vivo' graph structure found by measuring a single real-world process (*e.g.*, friend formation in a social network). To this end, GSL is especially important in real-world settings where the observed graph structure might be noisy, incomplete, or even unavailable (Halcrow et al., 2020).

In this paper, we develop the first unified framework for Graph Structure Learning, which we call Unified Graph Structure Learning (UGSL). We then provide a holistic benchmarking using our unified framework. The framework reformulates ten existing models into a single architecture, allowing for a comprehensive comparison of methods. We implement a wide range of existing models in our framework and conduct extensive analyses of the effectiveness of different components in the framework. Our results provide a clear and concise understanding of the different methods in this area as well as their strengths and weaknesses.

**Contributions.** Specifically, our contributions are the following: (1) **UGSL**, our unified framework for benchmarking GSL which encompasses over ten existing methods and four thousand different architectures in the same model. (2) **GSL benchmarking study**, the results of our GSL Benchmarking study, a first-of-its-kind effort that compared over *four thousand architectures* across six different datasets in twenty-two different settings, giving insights into the general effectiveness of the compo-

nents and architectures[1]. (3) **Open source code**, We have uploaded our code in the supplementary material. We are committed to open-sourcing both our code and data upon the acceptance of our paper, allowing other researchers to reproduce our results, build on our work, and use our code to develop their own GSL models.

## 2 PRELIMINARIES

Lowercase letters (*e.g.*, $n$) denote scalars. Bold uppercase (*e.g.*, $\boldsymbol{A}$) denotes matrices. Calligraphic letters (*e.g.*, $\mathcal{X}$) denote sets. Sans-serif (*e.g.*, MyFunc) denotes functions. $\boldsymbol{I}$ is the identity matrix. For a matrix $\boldsymbol{M}$, we represent its $i^{\text{th}}$ row as $\boldsymbol{M}_i$ and the element at the $i^{\text{th}}$ row and $j^{\text{th}}$ column as $\boldsymbol{M}_{ij}$.

Further, $\odot$ denotes Hadamard product, $\circ$ denotes function composition, Cos is cosine similarity of the input vectors, $\sigma$ denotes element-wise non-linearity, $^\top$ a transposition operation, and $||$ is a concatenation operation. We let $|\mathcal{M}|$ represent the number of elements in $\mathcal{M}$ and $||\boldsymbol{M}||_\mathsf{F}$ indicate the Frobenius norm of matrix $\boldsymbol{M}$. Finally, $[n] = \{1, 2, \ldots, n\}$.

Let graph $\mathsf{G} = (\boldsymbol{X}, \boldsymbol{A})$ be a graph with $n$ nodes, feature matrix $\boldsymbol{X} \in \mathbb{R}^{n \times d}$, and adjacency matrix $\boldsymbol{A} \in \mathbb{R}^{n \times n}$. Let in-degree diagonal matrix $\overleftarrow{\boldsymbol{D}}$ with $\overleftarrow{\boldsymbol{D}}_{ii}$ counting the in-degrees of node $i \in [n]$, and $\overrightarrow{\boldsymbol{D}}_{ii}$ counting its out-degree. Let $\mathcal{G} = \mathcal{X} \times \mathcal{A}$ denote the space of graphs with $n$ nodes. Let $\mathsf{G}^{(0)} = (\boldsymbol{X}^{(0)}, \boldsymbol{A}^{(0)}) \in \mathcal{G}$ be an **input** graph with feature matrix $\boldsymbol{X}^{(0)} \in \mathcal{X} \subseteq \mathbb{R}^{n \times d_0}$ and adjacency matrix $\boldsymbol{A}^{(0)} \in \mathcal{A} \subseteq \mathbb{R}^{n \times n}$. In most-cases, $\boldsymbol{X}^{(0)} = \boldsymbol{X}$ (and $d_0 = d$).

The **Graph Structured Learning (GSL) problem** is defined as follows: *Given* $\mathsf{G}^{(0)}$ *and a task* $T$ *find an adjacency matrix* $\mathbf{A}$ *which provides the best graph inductive bias for* $T$. Our proposed method UGSL captures functions of the form: $\mathsf{f} : \mathcal{G} \to \mathcal{G}$, where $\mathsf{f}$ denotes a graph generator model. Specifically, we are interested in methods that (iteratively) output graph structures as:

$$\mathsf{G}^{(\ell)} = (\boldsymbol{X}^{(\ell)}, \boldsymbol{A}^{(\ell)}) = \mathsf{f}^{(\theta_\ell, \ell)}(\boldsymbol{X}^{(\ell-1)}, \boldsymbol{A}^{(\ell-1)}) \quad \text{for} \quad \ell \in [L] \quad \text{with} \quad \mathsf{f}^{(\theta)} = \mathsf{f}^{(\theta_L, L)} \circ \cdots \circ \mathsf{f}^{(\theta_1, 1)} \tag{1}$$

Here, $\mathsf{f}^{(\theta_\ell, \ell)}$ represents the $\ell$-th graph generator model with parameters $\theta_l$. A variety of methods fit this framework. A class of these methods does not process an adjacency matrix as input. As such, they can be written as $\mathsf{f}^{(\theta)} : \mathcal{X} \to \mathcal{X} \times \mathcal{A}$. Another class does not output node features, *i.e.*, $\mathsf{f}^{(\theta)} : \mathcal{X} \times \mathcal{A} \to \mathcal{A}$. Nonetheless, we only consider $\mathsf{f}$ with $\mathcal{A} \in \mathsf{range}(\mathsf{f})$. Specifically, methods that output an adjacency matrix.

It is worth stating that some classical methods *e.g.*, $k$-nearest neighbors ($k$NN) and partial SVD, fit this framework, both with $L = 1$. Specifically, one can define $\boldsymbol{X}^{(0)}$ as given features and $\mathsf{f}^{(1)}$ for $k$-NN to output adjacency matrix $\boldsymbol{A}^{(k\text{NN})} = \mathsf{f}^{(1)}(\boldsymbol{X}^{(0)})$ as

$$\boldsymbol{A}_{ij}^{(k\text{NN})} = 1 \iff j \in \mathsf{NearestNeighbors}_k(i), \tag{2}$$

where $\mathsf{NearestNeighbors}_k(i) \subseteq [n]^k$ is $k$-nearest neighbors to $i$, per a distance function, *e.g.*, Euclidean distance. For partial SVD, one can set $\boldsymbol{A}^{(0)} = \boldsymbol{A}$ and set:

$$\boldsymbol{A}^{(\text{PARTIALSVD})} = \mathsf{f}^{(1)}(\boldsymbol{A}^{(0)}) = \boldsymbol{U}_* \boldsymbol{V}_*^\top \tag{3}$$

$$\text{where} \quad \mathbf{U}_*, \boldsymbol{V}_* = \underset{\boldsymbol{U}, \boldsymbol{V}}{\arg \min} \left\| \left( \boldsymbol{U}\boldsymbol{V}^\top - \boldsymbol{A}^{(0)} \right) \odot \mathbf{1}[\boldsymbol{A}^{(0)} > 0] \right\|_\mathsf{F} \tag{4}$$

Beyond fitting classical algorithms into UGSL, we are interested in unifying modern algorithms that infer the adjacency matrix with complex processes, *e.g.*, via a deep neural network. Further, we are interested in methods where the adjacency matrix produced by $\mathsf{f}$ is utilized in a downstream model, which can be trained in a supervised, unsupervised, or self-supervised, end-to-end manner.

## 3 UGSL: A UNIFIED FRAMEWORK FOR GRAPH STRUCTURE LEARNING

The objective of this section is to present a comprehensive unified framework for models designed for graph structure learning. We first describe our proposed unified framework using UGSL layers, a

---

[1]This is in contrast and complementary to the concurrent work of Zhou et al. (2023) which compared existing models using the available software for each model.

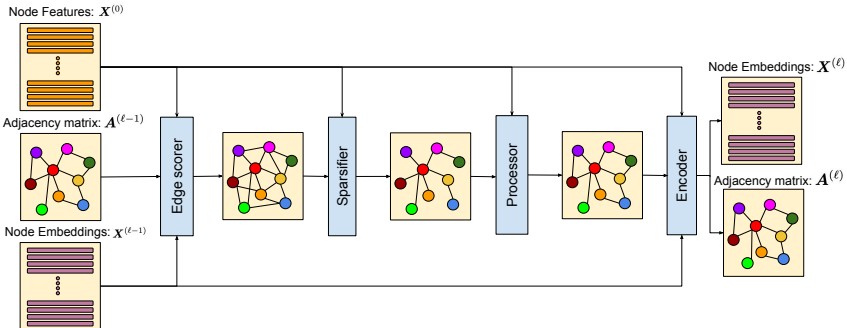

Figure 1: Overview of the $\ell$-th GSL layer, $\mathsf{f}^{(\ell,\theta^\ell)}$.

general layer for graph structure learning. The $\ell$-th UGSL layer defines function $\mathsf{f}^{(\ell,\theta^\ell)}$ as:

$$\mathsf{f}^{(\ell,\theta^\ell)} = \mathsf{Encoder}^{(\theta_\mathsf{E}^\ell)} \circ \mathsf{Processor}^{(\theta_\mathsf{P}^\ell)} \circ \mathsf{Sparsifier}^{(\theta_\mathsf{S}^\ell)} \circ \mathsf{EdgeScorer}^{(\theta_\mathsf{ES}^\ell)}, \tag{5}$$

the composition of 4 trainable modules. Multiple UGSL layers can be combined to create a UGSL model as in Equation 1. Next, we summarize the role of each module. Then, we show that many methods can be cast into the UGSL framework by specifying these modules (Table 15).

* Input: Each UGSL layer $\ell$ takes as input the output graph of the $(\ell - 1)$-th layer $\mathsf{G}^{(\ell-1)} = (\boldsymbol{X}^{(\ell-1)}, \boldsymbol{A}^{(\ell-1)})$, and the input graph $\mathsf{G}^{(0)} = (\boldsymbol{X}^{(0)}, \boldsymbol{A}^{(0)})$.

* EdgeScorer (w. parameters $\theta_\mathsf{ES}$) scores every[2] node-pair, producing output $\in \mathbb{R}^{n\times n}$.

$$\boldsymbol{A}^{(\mathsf{ES},\ell)} = \mathsf{EdgeScorer}(\mathsf{G}^{(0)}, \mathsf{G}^{(\ell-1)}; \theta_\mathsf{ES}) \tag{6}$$

* Sparsifier (w. parameters $\theta_\mathsf{S}$) Sparsifies the graph, *e.g.*, via top-$k$ or thresholding:

$$\boldsymbol{A}^{(\mathsf{S},\ell)} = \mathsf{Sparsifier}(\mathsf{G}^{(0)}, \mathsf{G}^{(\ell-1)}, \boldsymbol{A}^{(\mathsf{ES},\ell)}; \theta_\mathsf{S}) \tag{7}$$

* Processor (w. parameters $\theta_\mathsf{P}$) takes the output of the sparsifier and output a processed graph as:

$$\boldsymbol{A}^{(\mathsf{P},\ell)} = \mathsf{Processor}(\mathsf{G}^{(0)}, \mathsf{G}^{(\ell-1)}, \boldsymbol{A}^{(\mathsf{S},\ell)}; \theta_\mathsf{P}) \tag{8}$$

* Encoder (w. parameters $\theta_\mathsf{E}$) generates updated node embeddings $\boldsymbol{A}^{(\mathsf{E},\ell)}$ as:

$$\mathsf{G}^{(\ell)} = (\boldsymbol{X}^{(\ell)}, \boldsymbol{A}^{(\mathsf{E},\ell)}) = \mathsf{Encoder}(\mathsf{G}^{(0)}, \mathsf{G}^{(\ell-1)}, \boldsymbol{A}^{(\mathsf{P},\ell)}; \theta_\mathsf{E}) \tag{9}$$

* Output: The above modules are invoked for $\ell \in [L]$ giving final UGSL output of:

$$\mathsf{G}^{(L)} = (\boldsymbol{X}^{(L)}, \boldsymbol{A}^{(\mathsf{E},L)}). \tag{10}$$

## 4 EXPERIMENTAL DESIGN

We ablate a variety of choices for modules introduced in §3 over six datasets from various domains. Overall, we run hundreds of thousands of experiments, exploring module choices and hyperparameters. We conduct two kinds of explorations: line search (§4.1) and random search (§4.2).

**Datasets.** To benchmark using the UGSL framework, we experiment across six different datasets from various domains. The first class of datasets consists of the three established benchmarks in the GNN literature namely Cora, Citeseer, and Pubmed Sen et al. (2008). Another dataset is Amazon Photos (or Photos for brevity) Shchur et al. (2018) which is a segment of the Amazon co-purchase graph McAuley et al. (2015). The other two datasets are used extensively for classification with no

---

[2]Naive all-pairs require $\mathcal{O}(n^2)$ though can be approximated with hashing with $\mathcal{O}(n)$ resources.

| Positional encoding | Explanation |
|---|---|
| WL role | Positional encoding based on the Weisfeiler-Lehman absolute role (Shervashidze et al., 2011). |
| Spectral role | Positional encoding based on the top k eigenvectors of the graph laplacian. |

Table 1: An Overview of Different Positional Encodings.

structure known in advance. One is an image classification dataset Imagenet-20 (or Imagenet for brevity), a subset of Imagenet containing only 20 classes (totaling 10,000 train examples). We used a pre-trained Vision Transformer feature extractor (Dosovitskiy et al., 2021). The last dataset is a text classification dataset Stackoverflow Xu et al. (2017). We used a subset collected by Xu et al. (2017) consisting of 20,000 question titles associated with 20 categories. We obtained their node features from a pre-trained Bert Devlin et al. (2018b) model loaded from TensorFlow Hub.

**Implementation.** The framework and all components are implemented in Tensorflow (Abadi et al., 2015) using the TF-GNN library (Ferludin et al., 2022) for learning with graphs.

### 4.1 EXPERIMENTING WITH COMPONENTS OF UGSL

This section explores the key components of different UGSL modules and experiments with them.

**Base model.** A base model is a UGSL model that is used as a reference for comparisons. The objective of such a model is to be minimal and have popular components. Inspired by dual encoder models for retrieval Gillick et al. (2018), our base model uses only raw features in the input, has an MLP as edge scorer, k-nearest neighbors as sparsifier, no processor, and a GCN as encoder with no regularizer or unsupervised loss function. The base model is trained with a supervised classification loss. The components of the base model will be explained as we explore different options.

In the rest of this section, we explain different components of the framework and experiment with them. For analyzing different components, we consider the base model and only change the corresponding component to be able to measure the effectiveness of one component at a time in isolation.

**Input.** To make our proposed framework applicable to a wide range of applications and conduct coherent experiments on multiple datasets, we assume $A^{(0)}$ is an empty matrix. When a non-empty $A^{(0)}$ is required (*e.g.*, for computing positional encodings), we create a kNN graph from the raw features $X$. For $X^{(0)}$, many models assume $X^{(0)} = X$, *i.e.*, the input embeddings are simply the given node features. However, in some architectures, $X^{(0)}$ is defined differently to contain information from the input adjacency matrix $A^{(0)}$ too. Here, we explore two approaches one based on positional encodings of the Weisfeiler-Lehman (WL) absolute role of the nodes as proposed in Graph-Bert Zhang et al. (2020) and another based on positional encodings of top k eigenvectors of the graph Laplacian as proposed in Graph Transformer (Dwivedi and Bresson, 2020) following Dwivedi et al. (2020). We explain the two variants in Table 1 and compare their results in Section 4.1 (Input rows).

**Insight 1:** Adding information about the input adjacency matrix to the raw features yields improved performance compared to using raw features, across the five datasets. This holds even on datasets with less expressive raw features, *e.g.*, bag-of-words in Cora and Citeseer.

**Edge scorer.** Most edge scorers in the literature are variants of one of the following: Full parameterization (FP), Attentive (ATT), and multilayer perceptron (MLP). Table 3 shows a summary of edge scorers. Section 4.1 (Edge scorer rows) compares the edge scorers across the datasets.

**Insight 2:** The FP edge scorer outperforms MLP and ATT across the five datasets. The only dataset with ATT outperforming the two others is Imagenet. Imagenet uses a pre-trained Vision Transformer feature extractor (Dosovitskiy et al., 2021) and thus has more expressive features compared to the bag of words in Cora and Citeseer and Bert Devlin et al. (2018a) embeddings in Stackoverflow.

**Insight 3:** For FP, we tried two methods for initializing the adjacency: glorot-uniform Glorot and Bengio (2010), or initializing each edge based on the cosine similarity of the input features of the two nodes. We randomly searched over this hyperparameter (similar to the rest of the hyperparameters). We observed that the cosine-similarity initialization outperforms glorot-uniform.

|  | | Cora | | Citeseer | | Pubmed | | Photo | | Imagenet | | Stackoverflow | |
|---|---|---|---|---|---|---|---|---|---|---|---|---|---|
|  | **Component** | **Val** | **Test** | **Val** | **Test** | **Val** | **Test** | **Val** | **Test** | **Val** | **Test** | **Val** | **Test** |
| Input | Raw features | 68.20 | 65.30 | 69.20 | 67.30 | **72.60** | **69.60** | 80.91 | 79.87 | 96.80 | 94.25 | 76.28 | 75.86 |
|  | WL Roles | 50.60 | 49.60 | 38.60 | 29.90 | 60.80 | 58.50 | 79.87 | 78.76 | 96.70 | **94.35** | 68.82 | 69.51 |
|  | Spectral Roles | **69.60** | **68.10** | **69.50** | **68.10** | 68.40 | 67.90 | **83.40** | **81.26** | **96.85** | 93.55 | 76.08 | **76.21** |
| Edge scorer | MLP | 68.20 | 65.30 | 69.20 | 67.30 | **72.60** | **69.60** | 80.91 | 79.87 | 96.80 | 94.25 | 76.28 | 75.86 |
|  | ATT | 56.00 | 53.30 | 56.60 | 54.40 | 68.40 | 66.50 | 64.70 | 63.37 | 96.60 | **95.06** | 61.66 | 60.21 |
|  | FP | **69.60** | 67.70 | **71.40** | **69.90** | 71.80 | **72.20** | **86.80** | **85.80** | **96.90** | 94.25 | **76.73** | **77.72** |
| Sparsifier | kNN | 68.20 | 65.30 | 69.20 | **67.30** | 72.60 | 69.60 | 80.91 | 79.87 | 96.80 | 94.25 | 76.28 | 75.86 |
|  | εNN | 66.20 | 65.20 | 64.40 | 60.60 | 67.80 | 66.00 | 79.87 | 79.12 | 78.58 | 74.60 | 66.37 | 65.98 |
|  | d-kNN | 68.60 | **68.00** | 66.40 | 66.20 | **73.80** | 70.10 | **86.14** | **84.36** | 96.75 | 93.34 | **76.93** | 77.12 |
|  | Random d-kNN | 68.80 | 65.00 | 66.80 | 66.20 | 72.80 | **71.40** | 82.75 | 82.52 | **96.90** | **94.46** | 77.03 | **77.34** |
|  | Bernoulli relaxation | **70.20** | 45.40 | **71.20** | 64.10 | 66.40 | 41.30 | 46.80 | 45.78 | 91.06 | 31.65 | 57.55 | 49.00 |
| Processor | none | 68.20 | 65.30 | **69.20** | **67.30** | 72.60 | 69.60 | 80.91 | 79.87 | 96.80 | 94.25 | 76.28 | 75.86 |
|  | symmetrize | 67.40 | 66.80 | 68.80 | 65.90 | **73.60** | **71.80** | 88.50 | 86.80 | 96.75 | 94.96 | 74.17 | 74.81 |
|  | activation | 67.80 | 66.10 | 68.60 | 63.00 | 70.80 | 69.00 | 82.09 | 81.83 | 96.75 | 94.96 | 76.18 | 75.92 |
|  | activation-symmetrize | **68.41** | **67.12** | 68.23 | 62.30 | 73.20 | 70.30 | 86.67 | **86.93** | 96.75 | 95.26 | **77.63** | **77.83** |
| Encoder | GCN | **68.20** | **65.30** | **69.20** | **67.30** | **72.60** | 69.60 | 80.91 | 79.87 | 96.80 | 94.25 | 76.28 | 75.86 |
|  | GIN | 63.80 | 60.20 | 63.40 | 62.01 | 72.00 | **71.90** | **89.41** | **88.10** | **96.90** | 94.05 | **77.83** | 77.73 |
|  | MLP | 54.60 | 49.60 | 51.80 | 50.70 | 71.60 | 71.10 | 89.15 | 87.16 | **96.90** | **94.35** | 77.23 | **78.37** |
| Regularizer | none | 68.20 | 65.30 | 69.20 | 67.30 | 72.60 | 69.60 | 80.91 | 79.87 | **96.80** | 94.25 | **76.28** | 75.86 |
|  | closeness | 72.00 | 68.00 | **71.60** | 66.30 | 71.20 | 70.60 | **87.19** | 85.00 | 96.60 | 93.55 | 70.82 | 71.56 |
|  | smoothness | 69.20 | 66.20 | 71.00 | 66.60 | 76.60 | 72.21 | 86.54 | 84.00 | 96.75 | **95.06** | 72.32 | 70.98 |
|  | sparse-connect | 69.40 | 67.50 | 71.00 | 67.40 | **80.60** | **76.40** | 86.80 | 85.93 | **96.80** | 94.96 | 75.42 | 75.03 |
|  | log-barrier | 67.80 | 66.90 | 61.60 | 56.00 | 70.40 | 66.40 | 40.39 | 40.36 | 96.75 | 94.25 | 71.30 | 72.30 |
|  | sparse-connect, log-barrier | **72.20** | **68.60** | 71.00 | **67.70** | 79.20 | 75.80 | 83.66 | 82.84 | 96.70 | 94.25 | 71.72 | 70.12 |
| Unsupervised loss | none | 68.20 | 65.30 | 69.20 | 67.30 | 72.60 | 69.60 | 80.91 | 79.87 | 96.8 | 94.25 | **76.28** | **75.86** |
|  | denoising loss | 72.61 | 70.21 | 70.22 | 65.71 | 67.64 | 64.80 | 78.30 | 77.57 | 96.7 | 94.05 | 69.87 | 70.12 |
|  | contrastive loss | **73.41** | **71.40** | **72.25** | **68.41** | **79.42** | **75.70** | **92.94** | **91.04** | **96.9** | **95.16** | 73.77 | 73.71 |
| One or per layer | one adjacency | **68.20** | **65.30** | **69.20** | **67.30** | **72.60** | 69.60 | **80.91** | **79.87** | **96.80** | 94.25 | **76.28** | **75.86** |
|  | per layer adjacency | 67.41 | 65.22 | 68.21 | 67.00 | 71.80 | **69.90** | 69.28 | 70.27 | 96.70 | **94.56** | 73.17 | 74.33 |

Table 2: Results comparing different components used in UGSL.

| Edge scorer | Formula | Description |
|---|---|---|
| FP | $$\boldsymbol{A}_{ij}^{(\text{ES},\ell)} = \boldsymbol{V}_{ij}^{(\text{ES})}$$ | Each possible edge in the graph has a separate parameter learned directly. Initialization $\boldsymbol{V}_{ij}^{(\text{ES})} = \text{Cos}(\boldsymbol{X}_i^{(0)}, \boldsymbol{X}_j^{(0)})$ (cosine similarity of features) is significantly better than random. |
| ATT | $\boldsymbol{A}_{ij}^{(\text{ES},\ell)} = \frac{1}{m}\sum_{p=1}^{m}\text{Cos}(\boldsymbol{X}_i^{(l-1)} \odot \boldsymbol{V}_p^{(\text{ES})}, \boldsymbol{X}_j^{(l-1)} \odot \boldsymbol{V}_p^{(\text{ES})})$ | Learning a multi-head version of a weighted cosine similarity. |
| MLP | $\boldsymbol{A}_{ij}^{(\text{ES},\ell)} = \text{Cos}(\text{MLP}(\boldsymbol{X}_i^{(l-1)}), \text{MLP}(\boldsymbol{X}_j^{(l-1)}))$ | A cosine similarity function on the output of an MLP model on the input. |

Table 3: An overview of different edge scorers.

**Sparsifier.** Sparsifiers take a dense graph generated by the edge scorer and remove some of the edges. This reduces the memory footprint to store the graph (*e.g.*, in GPU), as well as the computational cost to run the GNN. Here, we experiment with the following sparsifiers: k-nearest neighbors (kNN), dilated k-nearest neighbors (d-kNN), $\epsilon$-nearest neighbors ($\epsilon$NN), and the concrete relaxation Jang et al. (2016) of the Bernoulli distribution (Bernoulli). Table 4 summarizes these methods. The results comparing the sparsifiers are summarized in Section 4.1 (Sparsifier rows).

**Insight 4:** In the case of $\epsilon$NN, since the number of edges in the graph is not fixed (compared to kNN) and the weights are being learned, the number of edges whose weight surpasses $\epsilon$ may be large, and so running $\epsilon$NN on accelerators may cause an out-of-memory error. [3]

**Insight 5:** The relaxation of Bernoulli does not generalize well to the test set due to its large fluctuation of loss at train time.

**Insight 6:** The kNN variants outperform $\epsilon$NN on our datasets. Among the kNN variants, dilated kNN works better than the other variants on five datasets. This is likely because dilation increases the receptive field of each node, allowing it to capture information from a larger region of the graph. This can increase the local smoothness of the node representations by approximating a larger neighborhood, leading to better performance in the downstream task.

**Processor.** Many works in the literature have explored the use of different forms of processing on the output of sparsifiers. These processing techniques can be broadly classified into three categories: i. applying non-linearities on the edge weights, *e.g.*, to remove negative values, ii. symmetrizing the

---

[3]To avoid this issue, we tried $\epsilon$NN on an extensive memory setup.

| Sparsifier | Formula | Description |
|---|---|---|
| kNN | $\boldsymbol{A}_{ij}^{(\text{S},\ell)} = \begin{cases} \boldsymbol{A}_{ij}^{(\text{ES},\ell)} & j \in \{N(i)_0, N(i)_1, \ldots, N(i)_k\} \\ 0 & \text{otherwise} \end{cases}$ | Define $N(i) = \arg\text{sort}_j \boldsymbol{A}_{ij}^{(\text{ES},\ell)}$. Then, for each node, keep the top $k$ edges with the highest weights. |
| d-kNN | $\boldsymbol{A}_{ij}^{(\text{S},\ell)} = \begin{cases} \boldsymbol{A}_{ij}^{(\text{ES},\ell)} & j \in \{N(i)_0, N(i)_d, \ldots, N(i)_{(k-1)d}\} \\ 0 & \text{otherwise} \end{cases}$ | Dilated convolutions Yu and Koltun (2015) were introduced in the context of graph learning by Li et al. (2019) to build graphs for very deep graph networks. We adapt dilated nearest neighbor operator to GSL. d-kNN adds a dilation with the rate $d$ to kNN to increase the receptive field of each node. |
| $\epsilon$NN | $\boldsymbol{A}_{ij}^{(\text{S},\ell)} = \begin{cases} \boldsymbol{A}_{ij}^{(\text{ES},\ell)} & \boldsymbol{A}_{ij}^{(\text{ES},\ell)} > \epsilon \\ 0 & \text{otherwise} \end{cases}$ | Only keeping the edges with weights greater than $\epsilon$. |
| Bernoulli | $\tilde{\boldsymbol{A}}_{ij}^{(\text{S},\ell)} = \text{Sigmoid}\left( \frac{1}{t}\left( \log \frac{\boldsymbol{A}_{ij}^{(\text{ES},\ell)}}{1-\boldsymbol{A}_{ij}^{(\text{ES},\ell)}} + \log \frac{a}{1-a} \right) \right)$ | Applying a relaxation of Bernoulli. Following Sun et al. (2022), we only keep the edges with weights greater than $\epsilon$ ($\epsilon$NN). Alternatively, the regularizer proposed in Miao et al. (2022) can be used to encourage sparsity. |

Table 4: An overview of different sparsifiers.

| Processor | Formula | Description |
|---|---|---|
| symmetrize | $\boldsymbol{A}_{ij}^{(\text{P},\ell)} = \frac{\boldsymbol{A}_{ij}^{(\text{S},\ell)}+\boldsymbol{A}_{ji}^{(\text{S},\ell)}}{2}$ | Making a symmetric version from the output of sparsifier. |
| activation | $\boldsymbol{A}_{ij}^{(\text{P},\ell)} = \sigma(\boldsymbol{A}_{ij}^{(\text{S},\ell)})$ | A non-linear function $\sigma$ is applied on the output of sparsifier. |
| activation-symmetrize | $\boldsymbol{A}_{ij}^{(\text{P},\ell)} = \frac{\sigma(\boldsymbol{A}_{ij}^{(\text{S},\ell)})+\sigma(\boldsymbol{A}_{ji}^{(\text{S},\ell)})}{2}$ | First applying a non-linear transformation and then symmetrizing the output. |

Table 5: An overview of different processors.

output of sparsifiers, *i.e.*, adding edge $v_j \to v_i$ if $v_i \to v_j$ survived sparsification, and iii. applying both (i.) and (ii.). The processors we used are listed in Table 5 and the results of applying these processors are shown in Section 4.1 (Processor rows).

**Insight 7:** Both activation (i.) and symmetrization (ii.) help improve the results, and their combination (iii.) performs best on several of the datasets.

**Encoder.** In this work, we experiment with GCN Kipf and Welling (2017) and GIN Xu et al. (2019) as encoder layers that can incorporate the learned graphs. As a baseline, we also add an MLP model that ignores the generated graph and only uses the features to make predictions. The results comparing the encoders are shown in Section 4.1 (Encoder rows).

**Insight 8:** On some of the datasets, the GNN variants GCN and GIN outperform MLP by a large margin. In some other datasets, MLP performs on par with the UGSL models.

**Insight 9:** On datasets with a strong MLP baseline, GIN outperforms GCN, which shows the importance of self-loops in these classification tasks.

**Loss Functions and Regularizers.** In our framework, we experiment with a supervised classification loss, four different regularizers, and two unsupervised loss functions. See Table 6 for a summary

| Name | Formula | Description |
|---|---|---|
| Supervised loss | $\sum_i \text{CE}(\text{label}_i, \text{pred}_i)$ | The supervision from the classification task as a categorical cross-entropy (CE represents the cross-entropy loss and the sum is over labeled nodes). |
| Closeness | $\|\boldsymbol{A}^{(0)} - \boldsymbol{A}\|_F^2$ | Discourages deviating from the initial graph. |
| Smoothness | $\sum_{i,j} \boldsymbol{A}_{ij}\text{dist}(v_i, v_j)$ | Discourages connecting (or putting a high weight on) pairs of nodes with dissimilar initial features. |
| sparse-connect | $\|\boldsymbol{A}\|_F^2$ | Discourages large edge weights. |
| Log-Barrier | $-\boldsymbol{1}^T \log(\boldsymbol{A}\boldsymbol{1})$ | Discourages low-degree nodes, with an infinite penalty for singleton nodes. |
| Denoising Auto-Encoder | $\sum_{i,j \in \mathcal{F}} \text{CE}(\boldsymbol{X}_{ij}, \text{GNN}_{\text{DAE}}(\tilde{\boldsymbol{X}}, \boldsymbol{A}))$ | Selects a subset of the node features $\mathcal{F}$, adds noise to them to create $\tilde{\boldsymbol{X}}$, and then trains a separate GNN to denoise $\tilde{\boldsymbol{X}}$ based on the learned graph. |
| Contrastive | $\frac{1}{2n}\left( \sum_i^n \log \frac{\left(\exp(\text{sim}(\boldsymbol{X}_i^L, \boldsymbol{Y}_i^L)/\tau)\right)}{\sum_{j=1}^n \left(\exp(\text{sim}(\boldsymbol{X}_i^L, \boldsymbol{Y}_j^L)/\tau)\right)} + \log \frac{\left(\exp(\text{sim}(\boldsymbol{Y}_i^L, \boldsymbol{X}_i^L)/\tau)\right)}{\sum_{j=1}^n \left(\exp(\text{sim}(\boldsymbol{Y}_i^L, \boldsymbol{X}_j^L)/\tau)\right)} \right)$ | Let $\texttt{G}_1 = (\boldsymbol{X}, \boldsymbol{A})$ and $\texttt{G}_2 = (\boldsymbol{X}, \text{combine}(\boldsymbol{A}^{(0)}, \boldsymbol{A}))$ ($\boldsymbol{A}$ is the learned structure and $\boldsymbol{A}^{(0)}$ is the initial $-\boldsymbol{I}$ if no initial structure). The combine function is a slow-moving weighted sum of the original and the learned graphs. Then, Let $\tilde{\texttt{G}}_1$ and $\tilde{\texttt{G}}_2$ be variants of $\texttt{G}_1$ and $\texttt{G}_2$ with noise added to the graph and features. The two views are fed into a GNN followed by an MLP to obtain node features $\boldsymbol{X}^{(L)}$ and $\boldsymbol{Y}^{(L)}$, on which the loss is computed. |

Table 6: A summary of loss functions (supervised and unsupervised) and regularizers.

of descriptions. Section 4.1 (Regularizer and Unsupervised loss rows) compares the regularizers and the two unsupervised losses in the UGSL framework respectively. Other forms of loss functions and regularizers have been tried in GSL. For instance, Wang et al. (2021) proposes homophily-enhancing objectives to increase homophily in a given graph or Jin et al. (2020) propose to use the nuclear norm of the learned adjacency as a regularizer to discourage higher ranks.

**Insight 10:** The log-barrier regularizer alone does not achieve effective results and this is mainly because this regularizer alone only encourages a denser adjacency matrix.

**Insight 11:** The sparse-connect regularizer outperforms the rest of the regularizers by a small margin on almost all datasets.

**Insight 12:** The unsupervised losses increase the performance of the base model the most. This might mainly be because of the supervision starvation problem studied in the GSL literature Fatemi et al. (2021). The contrastive loss is the most effective across most of the datasets.

**Learning an adjacency per layer.** In this experiment, we assessed two experimental UGSL layer setups. The first setup assumed a single adjacency matrix to be learned across different UGSL encoder layers. In the second setup, each UGSL encoder layer had a different adjacency matrix to be learned. The results from this experiment are in Section 4.1 (One or per layer rows).

**Insight 13:** Learning different adjacency matrices in each layer does not improve the performance. This finding suggests that the increase in model complexity outweighed the benefits of learning a separate adjacency matrix for each layer.

## 4.2 RANDOM SEARCH OVER ALL COMPONENTS

After gaining insights from the component analyses in Section 4.1, we excluded a few options and performed a random search over all remaining components of the UGSL framework, including their combinations and hyperparameters. We excluded $\epsilon$NN and Bernoulli because they require extensive memory and cannot be run on accelerators. For each dataset, we ran 30,000 trials.

**Best results obtained.** The trials with the best validation accuracy (val accuracy) are reported in Table 9, along with the corresponding test accuracy and the components in the architecture of the corresponding model. The results show that combining different components from different models further improves the base model. As we are running many trials for each dataset, we excluded the sparsifiers with $\epsilon$ variants to be able to run all trials on accelerators.

**Insight 14:** There is no single best-performing architecture; the best-performing components vary across datasets. This is because different datasets have different characteristics, such as the types of nodes and the features available for those. The fact that there is no single best-performing component suggests that it is important to carefully select the components that are most appropriate for the specific dataset and task at hand.

The best-performing components vary across datasets. However, there are some general trends:

**Insight 15:** While positional encodings were useful with a simple base architecture, they lost their effectiveness and did not provide additional improvement when using better architectures following an exhaustive search.

**Insight 16:** The ATT edge scorer is accompanied by a GIN encoder in the architectures where it participated. This further confirms Insight 2 on the ATT edge scorer working better for datasets with more expressive features as GIN better incorporates the self-loop information compared to GCN.

### 4.2.1 AVERAGE OVER THE BEST PERFORMING TRIALS FOR EACH COMPONENT

To gain more insights into the effectiveness of each component in the UGSL framework, we first identified the top-performing trials for each component on each dataset. We then computed the average accuracy across all datasets for each component. The results of this experiment are in Tables 7 and 8. For future tasks and datasets, this shows how probable it is for a component to be useful. For instance, d-kNN sparsifier, denoising loss, and contrastive loss achieved an average accuracy of 81.01%, 81.04%, and 80.99% respectively across all datasets.

**Insight 17:** D-kNN, denoising loss, and contrastive loss are more likely to be effective for a variety of tasks and datasets.

| WL | Spectral | MLP | FP | ATT | kNN | d-kNN | none | symmetrize | activation | activation-symmetrize |
|---|---|---|---|---|---|---|---|---|---|---|
| 78.58 | 80.56 | 80.53 | 80.92 | 79.90 | 80.71 | 81.01 | 80.24 | 80.66 | 80.77 | 80.54 |

Table 7: Average over the Best Performing Trials for Each Component.

| GCN | GIN | closeness | smoothness | sparseconnect | logbarrier | denoising | contrastive |
|---|---|---|---|---|---|---|---|
| 80.86 | 80.47 | 80.55 | 80.64 | 80.60 | 80.66 | 81.04 | 80.99 |

Table 8: Average over the Best Performing Trials for Each Component.

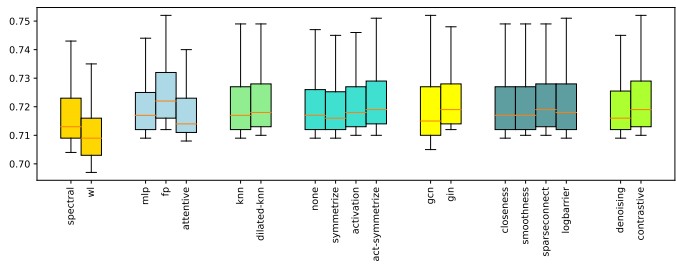

Figure 2: Results of the top 5% performing UGSL models in a random search on Pubmed.

**Top-performing components.** To get more insights from the different trials, in addition to the best results reported above, we analyze the top 5% performing trials for each component and visualize their results. Figure 2 show the box charts for different components across Pubmed and Imagenet (the rest in the appendix).

**Insight 18:** Tuning the choice of components in the final architecture is most important for edge scorer, followed by processor, unsupervised loss, and encoder. Tuning the regularizer and the positional encoding has a relatively lower effect.

**Best overall architectures.** In this experiment, we iterate over all architectures that were among those selected in the random search on all six datasets. For each dataset, we pick the best results corresponding to that architecture and then compute the average of the best results over all datasets. Table 10 shows the architectures corresponding to the top five test accuracy. This result brings insights into what architectures to explore for future applications of GSL.

**Insight 19:** The architectures listed in Table 10 mostly use both regularizers and unsupervised losses, which suggests that both are necessary to learn a graph structure.

**Insight 20:** GIN encoders participated in most of the top architectures, showing that using more expressive GNN layers helps even when learning a graph structure at the same time as learning the supervised task.

**Insight 21:** ATT was not effective most of the time in the base model, but it participated in some of the best architectures, which confirms the importance of searching over all combinations.

**Analyses of the graph structures.** We analyze the learned graph structures by computing graph statistics used in GraphWorld (Palowitch et al., 2022) (full descriptions in the Appendix), hoping to

| Dataset | Val Accuracy | Test Accuracy | Input Features | Edge scorer | Sparsifier | Processor | Encoder | Regularizers | Unsupervised Losses |
|---|---|---|---|---|---|---|---|---|---|
| Cora (base) | 68.20 | 65.30 | features | MLP | kNN | none | GCN | none | none |
| Cora (best) | 70.80 | 72.30 | features | FP | d-kNN | activation | GCN | none | denoising and contrastive |
| Citeseer (base) | 69.20 | 67.30 | features | MLP | kNN | none | GCN | none | none |
| Citeseer (best) | 72.00 | 71.20 | features | FP | kNN | activation-sym | GCN | closeness | none |
| Pubmed (base) | 72.60 | 69.60 | features | MLP | kNN | none | GCN | none | none |
| Pubmed (best) | 80.80 | 76.00 | features | MLP | d-kNN | activation | GIN | sparse-connect | contrastive |
| Photo (base) | 80.81 | 79.87 | features | MLP | kNN | none | GCN | none | none |
| Photo (best) | 92.55 | 89.84 | spectral | ATT | d-kNN | activation | GIN | sparse-connect | denoising and contrastive |
| Imagenet (base) | 96.80 | 94.25 | features | MLP | kNN | none | GCN | none | none |
| Imagenet (best) | 96.95 | 94.25 | features | FP | kNN | activation | GIN | smoothness | none |
| Stackoverflow (base) | 76.28 | 75.86 | features | MLP | kNN | none | GCN | none | none |
| Stackoverflow (best) | 77.48 | 77.82 | features | ATT | kNN | none | GIN | sparse-connect | none |

Table 9: Comparison of best random search models vs. base models

| Test Accuracy | Input | Edge scorer | Sparsifier | Processor | Encoder | Regularizers | Unsupervised Losses |
|---|---|---|---|---|---|---|---|
| 75.99 | features | FP | kNN | none | GCN | closeness, sparse-connect | contrastive |
| 75.92 | features | FP | d-kNN | symmetrize | GIN | none | contrastive |
| 75.71 | features | FP | kNN | activation | GIN | none | contrastive |
| 75.63 | features | ATT | kNN | symmetrize | GIN | closeness, smoothness, log-barrier | denoising, contrastive |
| 75.60 | features | ATT | kNN | symmetrize | GIN | closeness, log-barrier | contrastive |

Table 10: Top five performing architectures over the six datasets.

find correlations between graph structures learned by GSL methods and their downstream quality. Surprisingly, we observe little to no correlation $|\rho| \leq 0.1$ to both validation and test set performance with most metrics. Two metrics did have a limited effect on downstream classification performance: the number of nodes that have degree 1 ($\rho \approx -0.15$) and the overall diameter of the graph ($\rho \approx 0.2 - 0.3$). Degree-one nodes make it difficult for GNNs to pass messages leading to poor performance, however, the case for the diameter of the graph is much less clear and warrants further investigation.

## 5  DISCUSSIONS AND FUTURE DIRECTIONS

Graph Structure Learning (GSL) is a rapidly evolving field with many promising future directions.

**Scalability.** GSL is a rapidly evolving field with many promising future directions. One GSL becomes intractable for large number of nodes, as the number of possible edges in a graph grows quadratically with the number of nodes. One family of approaches to this is to utilize approximate nearest neighbor (ANN) search. ANN can mine 'closest' pairs of points in time approximately linear in the number of nodes, where 'close' is traditionally defined as within some similarity threshold $\epsilon$ or within the top-k neighbors of each point. ANN is a well studied problem with a number of approaches developed over the years *e.g.*, (Halcrow et al., 2020; Carey et al., 2022). The main limitation to this approach is that the ANN search is not differentiable. Instead these approaches must rely on some heuristic to guide the ANN search. Another family of approaches attempt to scale up graph transformers (Shirzad et al., 2023; Wu et al., 2022; Yun et al., 2019; Dwivedi and Bresson, 2020). While these techniques have improved the scalability of graph transformers, these techniques have yet to be shown to be effective for large scale graphs. We consider adapting our framework to larger scale datasets to be future work.

**Analyses of the graph structures.** One promising future direction is to develop new graph statistics that are more informative about the downstream quality of GSL methods. This could be done by identifying graph properties that are particularly important for GNNs, such as the presence of high-degree nodes or the absence of degree-one nodes, as done for natural graphs by Li et al.. Another important direction for future research is to investigate the relationship between the diameter of the graph and downstream performance in more detail. It is not clear why the diameter of the graph has a limited effect on downstream classification performance, but this could be because GNNs can learn long-range dependencies even in graphs with large diameters.

**UGSL beyond node classification.** In this work, we primarily focus on the node classification problem, which is the most popular task in the GSL literature (with the exception of some works such as graph classification in Sun et al. (2022)). However, since the output of a GSL model is node embeddings, we can use loss functions other than node classification (*e.g.*, link prediction, graph classification, etc.) to guide the learning towards those tasks.

**Conclusion.** In this paper, we presented UGSL, a unified framework for benchmarking GSL. UGSL encompasses over ten existing methods and four-thousand architectures in the same model, making it the first of its kind. We also conducted a GSL benchmarking study, comparing different GSL architectures across six different datasets in twenty-two different settings. Our findings provide insights about which components performed best both in isolation and also in a large random search. Overall, our work provides a valuable resource for the GSL community. We hope that UGSL will serve as a benchmark for future research and that our findings will help researchers to design more effective GSL models.

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

# A APPENDIX

## A.1 DATASETS

We ablate various UGSL components and hyperparameters across six different datasets from various domains. The first class of datasets consists of the three established benchmarks in the GNN literature namely Cora, Citeseer, and Pubmed Sen et al. (2008). For these datasets, we only feed the node features to the models **without** their original graph structures (edges). We also experiment with Amazon Photos (or Photos for brevity) Shchur et al. (2018) which is a segment of the Amazon co-purchase graph McAuley et al. (2015) with nodes representing goods. We only used the node features for this dataset as well. The other two datasets are used extensively for classification with no structure known in advance. One is an Image classification dataset Imagenet-20 (or Imagenet for brevity), a subset of Imagenet containing only 20 classes (totaling 10,000 train examples). We used a pre-trained Vision Transformer feature extractor (Dosovitskiy et al., 2021). The last dataset is Stackoverflow which is commonly used for short-text clustering/classification tasks. Here, we used a subset collected by Xu et al. (2017). consisting of 20,000 question titles associated with 20 different categories obtained from a dataset released as part of a Kaggle challenge. We obtained their node features from a pre-trained Bert Devlin et al. (2018b) model loaded from TensorFlow Hub. The GitHub repository contains scripts for loading the datasets from their original sources and also computing their embedding using pre-trained models (for Imagenet and Stackoverflow). The statistics of datasets used in the experiments can be found in Table 11.

## A.2 IMPLEMENTATION DETAILS.

We implemented the framework with all its components in Tensorflow (Abadi et al., 2015), used the TF-GNN library (Ferludin et al., 2022) for learning with graphs operations, and used Adam Kingma and Ba (2014) as optimizer. We performed early stopping and hyperparameter tuning based on the accuracy on the validation set for all datasets.

We fixed the maximum number of epochs to 1,000. We use two layers of UGSL in all the experiments. For the experiments on component analyses, for each dataset and component, we run a search with 1,024 trials. For the random search experiments over all components, we run 30,000 trials for each dataset. We run all our experiments on a single P100 GPU in our internal cluster. For each trial, the hyperparameters are selected randomly from a predefined range (for float hyperparameters) or list (for discrete hyperparameters). The range provided for the learning rate and weight decay of the Adam optimizer are (1e-3, 1e-1) and (5e-4, 5e-2) respectively. All dropout rates and non-linearities are selected from (0.0, 75e-2) and [relu, tanh] respectively. The rest of the hyperparameters are component dependent and are explained in the corresponding section for the component below.

We have uploaded our code in the supplementary material.

## A.3 EXPERIMENTING WITH COMPONENTS OF UGSL

In this section, we report charts obtained from the component analyses. These results are summarized in the main paper in spider charts.

**Input.** Figure 3a shows the spider chart corresponding to the results for feeding raw features as input, incorporating WL roles, and Spectral roles.

Table 11: Dataset statistics.

| Dataset | Nodes | Edges | Classes | Features | Label rate |
|---|---|---|---|---|---|
| Cora | 2,708 | 10,858 | 7 | 1,433 | 0.052 |
| Citeseer | 3,327 | 9,464 | 6 | 3,703 | 0.036 |
| Pubmed | 19,717 | 88,676 | 3 | 500 | 0.003 |
| Amazon-electronics-photo | 7,650 | 143,663 | 8 | 745 | 0.1 |
| Imagenet:20 | 11,006 | 0 | 20 | 1,024 | 0.72 |
| Stackoverflow | 19,980 | 0 | 21 | 768 | 0.1 |

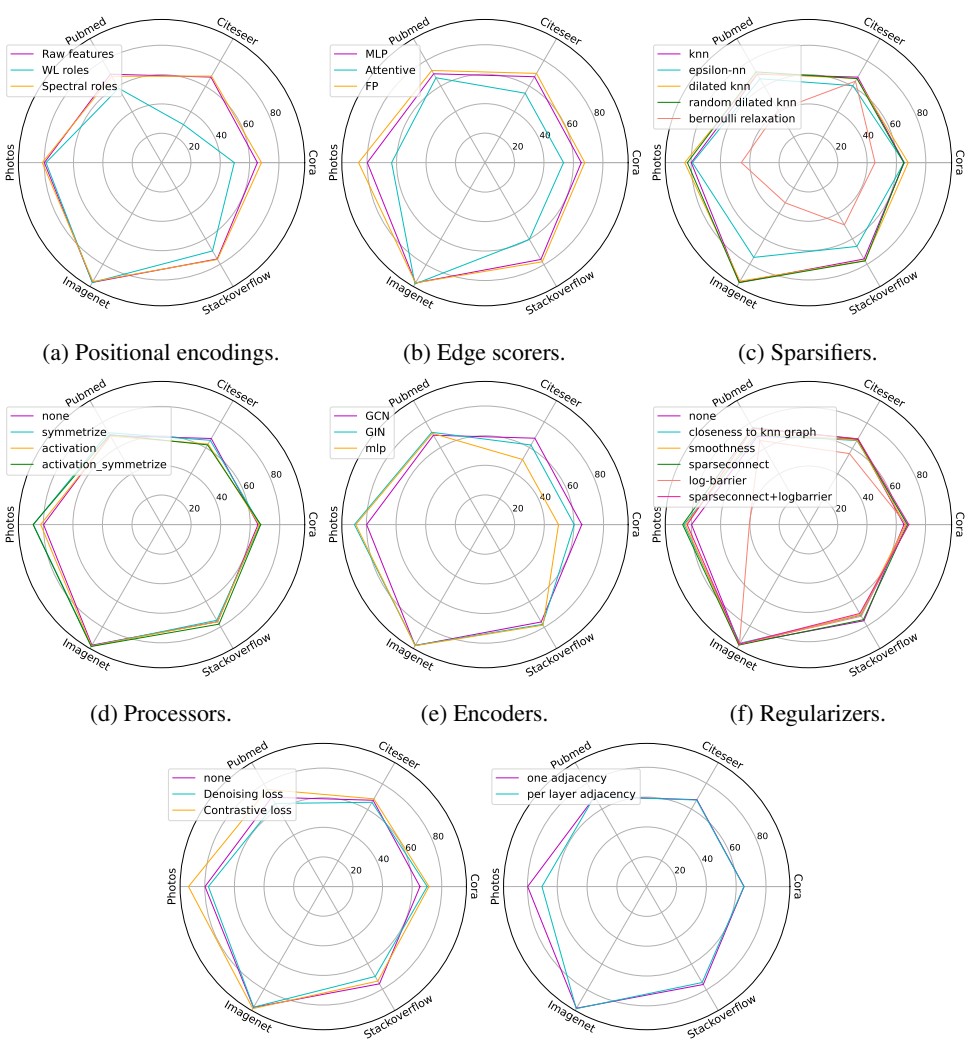

(a) Positional encodings.  (b) Edge scorers.  (c) Sparsifiers.

(d) Processors.  (e) Encoders.  (f) Regularizers.

(g) Unsupervised loss functions.  (h) One adjacency or per layer.

Figure 3: Line-search experiments summarizing thousands of runs. (a-d) ablate UGSL Modules, (e) ablates amending adjacency information into $X^{(0)}$, (f-g) ablate objective terms, and (h) ablates learning one adjacency or multiple. The raw results are in the appendix.

**Edge scorer.** For an MLP edge scorer, the number of layers is selected from the list [1, 2]. It has been mentioned by several recent papers (*e.g.*, Fatemi et al., 2021; Wu et al., 2023) in the area that initializing the MLP edge scorer such that it outputs the kNN graph has shown to be promising. For that, we selected hidden size and output size from [500, #features] (500 is changed to 250 for Pubmed with 500 features.). The layers in an MLP edge scorer are either initialized using glorot-uniform Glorot and Bengio (2010) or to an identity matrix (having layers of the size #features and initializing the weights as an identity will result in a kNN graph as initialization). The ATT edge scorer weights have been initialized either randomly or to ones (to replicate kNN in the initialization). Figure 3b shows the spider chart corresponding to the charts for comparing three edge scorers MLP, ATT, and FP.

**Sparsifier.** The sparsifiers that have kNN variants have their k selected from [15, 20, 25, 30]. d-kNN has a hyperparameter $d$ selected from [2, 3] and a boolean hyperparameter to do random dilation (random d-kNN) or not. The sparsifiers with a hyperparameter as $\epsilon$ have it selected from (0.0, 1.0). Figure 3c shows the results for multiple sparsifiers we tried in the architecture.

When using these sparsifiers while learning the parameters of the GSL model, following most of the successful work in this area, we only send gradients toward non-zero values in the output of the sparsifier (*e.g.*, a node being among the top k neighbors in the kNN sparsifier). As discussed in the straight-through estimator proposed by Bengio et al. (2013), this computes biased gradients, but it works well in practice because it enforces a prior that for a given input, most of the factors in the model are irrelevant and would be represented by zeros in the representation. Bengio et al. (2013) also investigated multiplying the gradient by the derivative of the sigmoid, but they found that better results were obtained without multiplying the derivative of the sigmoid.

**Processor.** Figure 3d shows the results for multiple processors we tried in the architecture. The only hyperparameters for the processors are the non-linearity function selected from [relu, tanh].

**Encoder.** The encoder has the number of hidden units as a hyperparameter selected from [16, 32, 64, 128] (non-linearity and dropout rate have been discussed above). Figure 3d shows the results for multiple processors we tried in the architecture.

**Loss functions and regularizers.** Each regularizer has its own weight selected from (0.0, 20.0). The hyperparameters for the denoising autoencoder and contrastive loss have been optimized based on the suggestions proposed in their corresponding paper. The denoising autoencoder has another GNN inside it with the number of hidden units selected from (512, 1024). The contrastive loss has a mask rate, temperature, and tau selected from (1e-2, 75e-2), (0.1, 1.0), and (0.0, 0.2) respectively. Figure 3f shows the results for multiple regularizers we tried in the architecture. Figure 3g shows the results for the unsupervised loss functions we tried in UGSL.

**Learning an adjacency per layer.** In this experiment, we assessed two experimental UGSL layer setups. The first setup assumed a single adjacency matrix to be learned across different UGSL encoder layers. In the second setup, each UGSL encoder layer had a different adjacency matrix to be learned. The results from this experiment are in Figure 3h.

## A.4 RANDOM SEARCH OVER ALL COMPONENTS

In this section, we provide more analyses for our extensive random search over all components.

### A.4.1 TOP-PERFORMING COMPONENTS

To get more insights from the different trials, in addition to the best results reported above, we analyze the top $5\%$ performing trials for each component and visualize their results. The box chart for Pubmed is in the main paper. Figures 4 to 8 show the box charts corresponding to Cora, Citeseer, Photo, Stackoverflow, and Imagenet datasets.

## A.5 ANALYSES OF THE GRAPH STRUCTURES

We study graph-level statistics of the constructed graphs using a collection of 7 metrics taken from GraphWorld Palowitch et al. (2022). Table 12 reports median statistics per dataset and Table 13 contrasts that with statistics of top-1% scoring graphs. We can observe that there are no significant

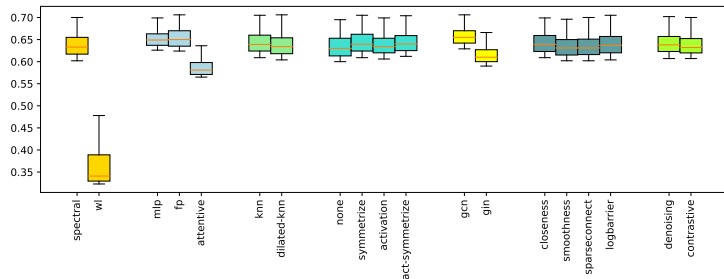

Figure 4: Results of the top 5% performing UGSL models in a random search on Cora.

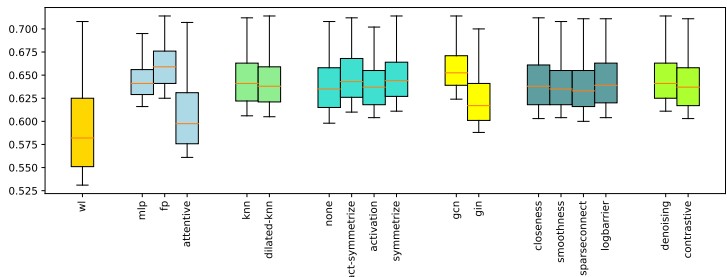

Figure 5: Results of the top 5% performing UGSL models in a random search on Citeseer.

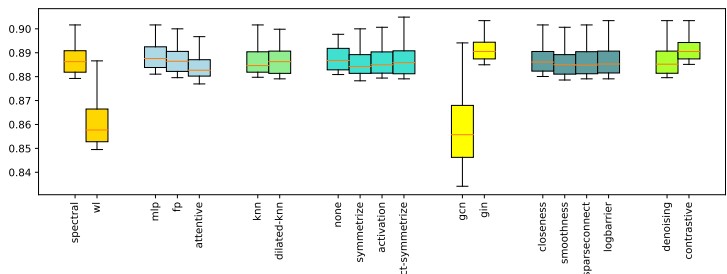

Figure 6: Results of the top 5% performing UGSL models in a random search on Photo.

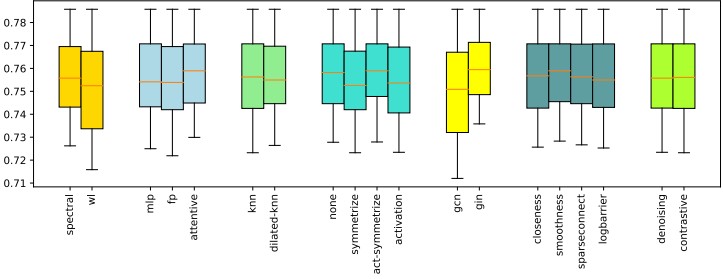

Figure 7: Results of the top 5% performing UGSL models in a random search on Stackoverflow.

differences across the best- and average-performing graphs except slightly lower clustering coefficient and higher diameter of best-performing graphs.

Additionally, we compute Spearman rank correlation to the test set accuracies, presented in Table 14. Here, we do not observe significant correlations except the ones reported in the main paper body.

Table 12: Median dataset statistics across all random search trials for all datasets. We report the average node degree $\bar{d}$, degree power law exponent $\alpha$, graph diameter, local ($\overline{CC_l}$) and global ($CC_g$) clustering coefficients, and graphs' spectral radius $\lambda_{\max}(A)$ and algebraic connectivity $\lambda_2(L)$.

| dataset | $\bar{d}$ | $\alpha$ | Diam. | $\overline{CC_l}$ | $CC_g$ | $\lambda_{\max}(A)$ | $\lambda_2(L)$ |
|---|---|---|---|---|---|---|---|
| Cora | 10.43 | 1.43 | 12 | 0.13 | 0.05 | 12.93 | 0.84 |
| Citeseer | 7.15 | 1.43 | 13 | 0.11 | 0.04 | 9.45 | 0.84 |
| Pubmed | 5.37 | 1.50 | 16 | 0.17 | 0.01 | 7.35 | 0.86 |
| Photo | 8.16 | 1.41 | 6 | 0.25 | 0.01 | 13.69 | 0.85 |
| Imagenet | 8.04 | 1.40 | 15 | 0.23 | 0.12 | 13.11 | 0.85 |
| Stackoverflow | 8.11 | 1.41 | 17 | 0.15 | 0.03 | 12.42 | 0.87 |

Table 13: Median dataset statistics for top 1% performing runs.

| dataset | $\bar{d}$ | $\alpha$ | Diam. | $\overline{CC_l}$ | $CC_g$ | $\lambda_{\max}(A)$ | $\lambda_2(L)$ |
|---|---|---|---|---|---|---|---|
| Cora | 10.94 | 1.41 | 13 | 0.10 | 0.09 | 13.74 | 0.84 |
| Citeseer | 7.90 | 1.41 | 13 | 0.06 | 0.06 | 9.27 | 0.84 |
| Pubmed | 5.28 | 1.48 | 17 | 0.01 | 0.00 | 6.04 | 0.85 |
| Photo | 8.01 | 1.43 | 9 | 0.03 | 0.02 | 11.93 | 0.85 |
| Imagenet | 9.11 | 1.39 | 16 | 0.22 | 0.14 | 14.22 | 0.85 |
| Stackoverflow | 8.17 | 1.46 | 19 | 0.13 | 0.04 | 12.08 | 0.87 |

Table 14: Spearman rank correlations between dataset statistics and test set accuracies.

| dataset | $\bar{d}$ | $\alpha$ | Diam. | $\overline{CC_l}$ | $CC_g$ | $\lambda_{\max}(A)$ | $\lambda_2(L)$ |
|---|---|---|---|---|---|---|---|
| Cora | 0.06 | 0.22 | 0.15 | 0.06 | 0.16 | 0.02 | -0.10 |
| Citeseer | 0.16 | 0.23 | 0.20 | -0.11 | 0.02 | 0.03 | -0.18 |
| Pubmed | -0.05 | 0.09 | -0.33 | 0.29 | 0.23 | 0.09 | 0.34 |
| Photo | -0.08 | 0.22 | -0.15 | -0.24 | 0.04 | -0.03 | 0.08 |
| Imagenet | 0.09 | -0.03 | 0.13 | 0.13 | 0.23 | 0.16 | 0.11 |
| Stackoverflow | -0.02 | 0.09 | 0.01 | 0.06 | -0.14 | -0.05 | 0.13 |

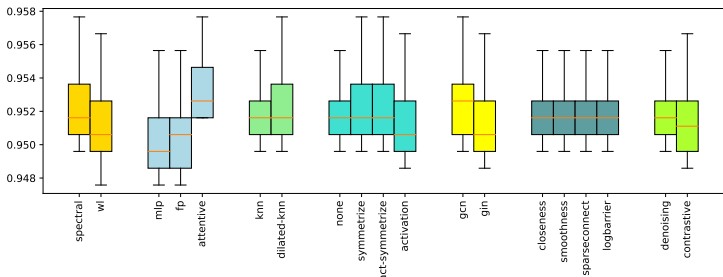

Figure 8: Results of the top 5% performing UGSL models in a random search on Imagenet.

| Model | Input | Edge scorer | Sparsifier | Processor | Regularizers | Unsupervised Losses |
|---|---|---|---|---|---|---|
| GLCN Jiang et al. (2019) | features | MLP | none | activation | sparse-connect, closeness, log-barrier | none |
| JLGCN Tang et al. (2019) | features | MLP | none | activation | smoothness | none |
| DGCNN Wang et al. (2019) | features | MLP | kNN | activation | none | none |
| LDS Franceschi et al. (2019) | features | FP | Bernoulli | none | none | none |
| IDGL Chen et al. (2020) | features | ATT | $\epsilon$NN | activation | sparse-connect, log-barrier | none |
| Graph-Bert Zhang et al. (2020) | features, WL and spectral | MLP | none | activation | none | none |
| GRCN Yu et al. (2021) | features | MLP | kNN | none | none | none |
| SLAPS Fatemi et al. (2021) | features | FP, MLP, ATT | kNN | activation-sym | none | denoising |
| SUBLIME Liu et al. (2022) | features | FP, MLP, ATT | kNN | activation-sym | none | contrastive |
| VIB-GSL Sun et al. (2022) | features | MLP | Bernoulli | none | none | denoising |

Table 15: Examples of existing models in UGSL along their components (all with a GCN encoder).

## A.6 EXAMPLES OF EXISTING GSL MODELS IN UGSL

Up to this point, we have explained the main components of UGSL. In Table 15, we describe some of the existing models along with their corresponding components in UGSL.

## A.7 FUTURE DIRECTIONS

**Scalability.** GSL becomes intractable for large number of nodes, as the number of possible edges in a graph grows quadratically with the number of nodes. One family of approaches to this is to utilize approximate nearest neighbor (ANN) search. ANN can mine 'closest' pairs of points in time approximately linear in the number of nodes, where 'close' is traditionally defined as within some similarity threshold $\epsilon$ or within the top-k neighbors of each point. ANN is a well studied problem with a number of approaches developed over the years *e.g.*, (Halcrow et al., 2020; Carey et al., 2022). The main limitation to this approach is that the ANN search is not differentiable. Instead these approaches must rely on some heuristic to guide the ANN search. Another family of approaches attempt to scale up graph transformers (Yun et al., 2019; Dwivedi and Bresson, 2020). Nodeformer (Wu et al., 2022) addresses the scalability issue by kernelizing its similarity function using random feature maps, reducing the complexity of the full message passing step from $\mathcal{O}(N^2)$ to $\mathcal{O}(N)$. It further addresses the associated issue of differentiability with the Gumbel-Softmax reparameterization trick. GraphGPS (Rampášek et al., 2022) uses a combination of a message passing neural network with a transformer and considers a few different sparse transformer options such as Performer (Choromanski et al., 2020). However it generally found that these options underperformed a standard transformer. Exphormer (Shirzad et al., 2023) adapts the framework from GraphGPS and combines local attention over a graph with global attention and attention over a random expander graph. While some techniques have improved the scalability of graph transformers, these techniques have yet to be shown to be effective for large scale graphs. We consider adapting our framework to larger scale datasets to be future work (Figure 9).

**Analyses of the graph structures.** One promising future direction is to develop new graph statistics that are more informative about the downstream quality of GSL methods. This could be done by identifying graph properties that are particularly important for GNNs, such as the presence of high-degree nodes or the absence of degree-one nodes, as done for natural graphs by Li et al.. Another important direction for future research is to investigate the relationship between the diameter of the graph and downstream performance in more detail. It is not clear why the diameter of the graph has a

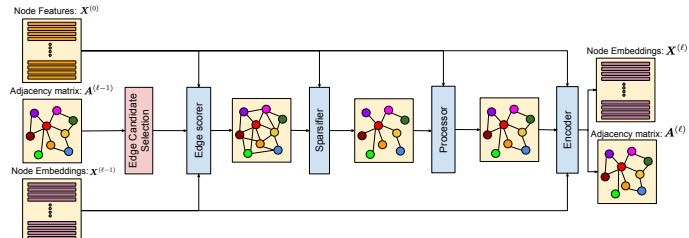

Figure 9: Overview of the $l$-th *scalable* GSL layer.

limited effect on downstream classification performance, but this could be because GNNs can learn long-range dependencies even in graphs with large diameters.

**UGSL beyond node classification.** In this work, we primarily focus on the node classification problem, which is the most popular task in the GSL literature (with the exception of some works such as graph classification in Sun et al. (2022)). However, since the output of a GSL model is node embeddings, we can use loss functions other than node classification (*e.g.*, link prediction, graph classification, etc.) to guide the learning towards those tasks.

