# OpenReview forum: "UGSL: A Unified Framework for Benchmarking Graph Structure Learning"
_ICLR.cc/2024/Conference — Submitted to ICLR 2024_

### Official Review · Reviewer_RW4a · 2023-10-28

**Soundness:** 2 fair
**Presentation:** 2 fair
**Contribution:** 1 poor
**Rating:** 3
**Confidence:** 4

**Summary:**

This paper presents a framework to unify and experimentally investigate existing graph structure learning methods. The unified framework consists of EdgeScorer, Sparsifier, Processor, Encoder and Output components. With experimental comparisons, it provides some interesting insights without serious proof.

**Strengths:**

It is interesting to provide a review and comparison to graph structure learning.

**Weaknesses:**

Although this may be the first work to present an experimental comparison of graph structure learning, my main concern is the contribution to the community of graphs. This paper seems like a review of this field. However, I don't think it is novel since most of the insights are without serious proof. Some conclusions may be correct with the limited methods.

**Questions:**

See weaknesses.

---

### Official Review · Reviewer_WpgZ · 2023-10-31

**Soundness:** 1 poor
**Presentation:** 3 good
**Contribution:** 2 fair
**Rating:** 5
**Confidence:** 4

**Summary:**

To tackle the inconsistent setups of various methods in the GSL domain, this paper presents a unified framework for evaluating GSL methods. This framework encompasses over 10 existing methods and four thousand diverse architectures. By conducting experiments on six datasets, valuable insights are provided regarding the efficacy of these components.

**Strengths:**

* A unified GSL framework is devised, which splits the GSL framework into four components, allowing for seamless substitution of different components.
* The author conducted over 30,000 comparisons on each dataset and offered numerous valuable insights.

**Weaknesses:**

* The scope of the work is limited. The authors only utilize node features on all datasets, without using their original graph structures. This may limit the scope of the benchmark, especially when some GSL methods are specifically designed for refining graphs[2, 3, 5]. Moreover, only node classification is included. Other tasks such as graph-level tasks should also be taken into consideration. It would also be better to compare the time complexity and memory consumption of different components.
* I'm concerned about whether UGSL framework can cover all the methods in Table 15. It seems that certain key designs of typical methods, such as the Bilevel Programming of LDS[1] and the Iteration Mechanism of IDGL[2], may not be adequately represented in UGSL. Additionally, common postprocessing techniques in GSL work, such as residual connection[2, 3, 4], are missing in the paper.
* Regarding the experiments,  variance is not given in many results, which is necessary to draw statistically significant conclusions regarding the relative merits of different components. In Table 7, 8, 10, it would be better to use the average ranking rather than average accuracy due to the potential large numerical variations of accuracy across different datasets. Lastly, while this may be beyond the scope, it would be interesting to look into the preferences of different dataset characteristics for specific components.

[1] Franceschi, Luca, et al. "Learning discrete structures for graph neural networks." *International conference on machine learning*. ICML, 2019.

[2] Chen, Yu, Lingfei Wu, and Mohammed Zaki. "Iterative deep graph learning for graph neural networks: Better and robust node embeddings." NeurIPS, 2020

[3] Yu, Donghan, et al. "Graph-revised convolutional network." ECML PKDD, 2020

[4] Zhao, Jianan, et al. "Heterogeneous graph structure learning for graph neural networks." AAAI, 2021.

[5] Liu, Yixin, et al. "Towards unsupervised deep graph structure learning." WWW, 2022.

**Questions:**

This paper seems easy to reproduce, though the data splits are not given. Besides, the results supporting Insight 3,4,5 are missing. Please make sure that all insights are supported by corresponding tables or figures. The experimental settings, results and insights should be more detailed and organized.

---

### Official Review · Reviewer_k3xx · 2023-10-31

**Soundness:** 2 fair
**Presentation:** 3 good
**Contribution:** 2 fair
**Rating:** 3
**Confidence:** 4

**Summary:**

In this paper, the authors focus on the growing area of graph neural networks (GNNs) and their applications. While traditional GNN approaches assume a predefined graph structure, recent methods have expanded GNN applicability by demonstrating their effectiveness even in the absence of an explicitly provided graph structure. Instead, these methods learn both the GNN parameters and the graph structure simultaneously. The challenge arises from the diverse experimentation setups used in previous studies, making it hard to compare their effectiveness directly. To address this issue, the authors introduce a benchmarking strategy called Unified Graph Structure Learning (UGSL). UGSL reformulates existing models into a unified framework, enabling the implementation of a wide range of methods. Through extensive analyses, the authors evaluate the effectiveness of different components within the framework. Their results offer a comprehensive understanding of various methods in this domain, elucidating their respective strengths and weaknesses. This research provides valuable insights into the complex landscape of graph structure learning and offers a standardized approach for comparing different techniques in the field of graph neural networks.

**Strengths:**

- This paper systematically studies the problem of graph structure learning and compares different types of methods on different benchmark graphs.

- Several insights have been provided based on the benchmarking studies which are potentially useful when selecting graph structure learning models
.

**Weaknesses:**

- These insights provided in this paper rely on limited graph benchmark datasets and these datasets have limitations in 1) they are in relatively small sizes, e.g., 10k nodes. 2) Datasets with graphs reflect more homophily but heterophily. However, in practice, many graphs show the pattern of a mixture of both homophily and heterophily. 3) The number of datasets is relatively small, e.g., there are only 3 datasets w/ and w/o graphs respectively. 4) All the tasks are node classification (although this has been discussed in future directions). It is necessary to consider different tasks in order to make more convincing conclusions.

- In addition to graph datasets, the tested base models are limited to GCN and GIN. More types of GNNs should be compared in order to make more convincing conclusions and/or more general insights.

- To show the effectiveness of structure learning, more insights from connecting the learned structures and downstream tasks should be discussed.

**Questions:**

To make more convincing conclusions and/or more general insights, the benchmarking study should be more extensive including more types of GNNs, e.g., base models, and more graph data (with different sizes, characteristics, and tasks). Therefore, my questions include:

- Will the insights/conclusions be consistent with different GNNs and different datasets?
- Will the insights/conclusions be consistent with different downstream tasks?

Besides, it will be interesting to have a more detailed discussion on the relationships between the learned graph structures and downstream tasks.

---

### Official Review · Reviewer_B447 · 2023-10-31

**Soundness:** 2 fair
**Presentation:** 2 fair
**Contribution:** 2 fair
**Rating:** 3
**Confidence:** 4

**Summary:**

This paper talks about a new way to test graph structure learning (GSL) methods. The authors look at different parts and designs in this method, see how they work, and tell us what's good and bad about current GSL methods. They also talk about UGSL layers and show how other models can fit into their new method. The paper has many tests on different data and looks at more than 4,000 designs.

**Strengths:**

1. The authors compare many structures, and their new method is clear.
2. Some ideas in the paper are helpful and can make people think more about this topic.

**Weaknesses:**

1. Lack of the baselines and large graph datasets: though the author did great jobs in searching on various architectures, the work itself is not enough for supporting a comprehensive paper. The used graph is relatively small.
2. Lack of the novelty: divide the training process of GSL is trivial, and and some tests are hard to understand because they are too similar.

**Questions:**

1. In Insight 16, you said "GIN is better than GCN for self-loop information." Where did you get this idea? Please show where you found this.
2. The paper is hard to read, especially the tests. Maybe you can use pictures or charts to show the differences in each test section.
3. Can you add more basic examples and use bigger graphs in your tables?

---

### Meta-Review · Area_Chair_idaw · 2023-12-06

**Metareview:**

This paper proposes a benchmarking strategy for graph structure learning using a unified framework, named as Graph Structure Learning (UGSL). This framework encompasses over 10 existing methods and four thousand diverse architectures. Experiments on six datasets were conducted to validate the efficacy of these components.

This topic is an interesting one. The unified GSL framework can allows for seamless substitution of different components. Several insights are potentially useful for selecting graph structure learning models.

However, the novelty of this paper is limited, and it can not cover those GSL method for refining graphs. More experiments should be conducted on large graph datasets, and more different GNN models. Finally, the paper should be polished more carefully, such as adding the test variance.

**Justification For Why Not Higher Score:**

The novelty is limited.

**Justification For Why Not Lower Score:**

N/A

---

### Decision · Program_Chairs · 2024-01-16

Reject